# The Role of Saponins in the Treatment of Neuropathic Pain

**DOI:** 10.3390/molecules27123956

**Published:** 2022-06-20

**Authors:** Bei Tan, Xueqing Wu, Jie Yu, Zhong Chen

**Affiliations:** 1Key Laboratory of Neuropharmacology and Translational Medicine of Zhejiang Province, School of Pharmaceutical Sciences, Zhejiang Chinese Medical University, Hangzhou 310053, China; iris813704@163.com (B.T.); wxq5220@126.com (X.W.); yujie@zcmu.edu.cn (J.Y.); 2School of Basic Medical Science, Zhejiang Chinese Medical University, Hangzhou 310053, China

**Keywords:** neuropathic pain, saponins, chemical structure, mechanism

## Abstract

Neuropathic pain is a chronic pain caused by tissue injury or disease involving the somatosensory nervous system, which seriously affects the patient’s body function and quality of life. At present, most clinical medications for the treatment of neuropathic pain, including antidepressants, antiepileptic drugs, or analgesics, often have limited efficacy and non-negligible side effects. As a bioactive and therapeutic component extracted from Chinese herbal medicine, the role of the effective compounds in the prevention and treatment of neuropathic pain have gradually become a research focus to explore new analgesics. Notably, saponins have shown analgesic effects in a large number of animal models. In this review, we summarized the most updated information of saponins, related to their analgesic effects in neuropathic pain, and the recent progress on the research of therapeutic targets and the potential mechanisms. Furthermore, we put up with some perspectives on future investigation to reveal the precise role of saponins in neuropathic pain.

## 1. Introduction

Pain was defined in 2020 by the International Association for the Study of Pain (IASP) as “an unpleasant sensory and emotional experience associated with, or resembling that associated with, actual or potential tissue damage”, which is divided into acute pain and chronic pain [1]. Under physiological conditions, acute pain is the body’s nociception that protects the organization from further injury of noxious stimuli by triggering the body’s reflex defense behavior, which plays a positive role in ensuring individual safety and health. However, chronic pain still exists for a long time after injury repairment, and has lost the significance of protecting individuals. It makes patients feel uncomfortable or painful, resulting in many physiological and psychological disorders that greatly affect the quality of life of patients [2]. Neuropathic pain is one of the chronic pains, and is caused by somatic sensory nervous system injury or disease [3,4]. The incidence of neuropathic pain in the population is as high as 7–10% [5]. The main features include spontaneous pain, unpleasant abnormal sensation (paresthesia), enhanced response to pain stimuli (hyperalgesia), and pain to stimuli that usually do not cause pain (allodynia) [3]. Neuropathic pain often causes comorbidity of emotional disorders. A total of 50–60% of patients have anxiety disorder and 36–71% have depression disorder, which seriously affect the physical and mental health and quality of life of patients [6,7]. Classical analgesics have poor therapeutic effects on neuropathic pain. At present, the recommended therapeutic drugs mainly include tricyclic antidepressants, serotonin and norepinephrine reuptake inhibitors, pregabalin, and gabapentin, but these drugs are only effective for about 50% of patients and have limited pain relief. Moreover, long-term administration will cause many intolerable side effects [8,9]. Therefore, the clinical treatment of neuropathic pain still faces great challenges.

Recently, the effective components, extracted from Chinese herbal medicine, for the prevention and treatment of neuropathic pain have gradually become the research focus to explore new analgesics. Saponins are widely distributed in nature, including fungi, ferns, plants, animals, and marine life [10]. The word “saponins” is a free translation from the English name Saponin, which is derived from the Latin “*Sapo*”, which means soap. In the chemical structure of saponins, due to the lipophilicity of the aglycone to different degrees, the sugar chain has strong hydrophilicity, and the aqueous solution can produce persistent, soap-like foam after shaking. Saponins have a wide range of physiological activities. A large body of studies have shown that saponins extracted from many Chinese herbal medicines have potential analgesic effects in various neuropathic pain models, which brought great potential for the development of new analgesic drugs. Therefore, we conducted a literature review to examine the role of saponins in the treatment of neuropathic pain. The search was performed in the PubMed and CNKI databases with the following keywords: saponins and neuropathic pain or pain. Relevant articles, including research papers, reviews, and their selective references, were examined systematically and are cited in the present review. This review focuses on the research progress of saponins in neuropathic pain. In addition, the chemical structure, mechanism of action, and experimental models of their biologically active ingredients are also introduced and discussed.

## 2. The Chemical Properties of Saponins

Saponins are a class of compounds with diverse structures, consisting of sapogenin and glycosyl groups [11]. The common ones of the saccharides that make up saponins are D-glucose, D-galactose, D-xylose, L-arabinose, and L-rhamnose, etc. According to the different structures of sapogenin, saponins are usually divided into two categories: triterpene saponins and steroidal saponins.

The triterpene aglycon structure of triterpene saponins is different. According to the basic carbon skeleton of the aglycon, triterpenoids can be divided into chain triterpenes, monocyclic triterpenes, bicyclic triterpenes, tricyclic triterpenes, tetracyclic triterpenes, and pentacyclic triterpenes. In nature, plants are dominated by tetracyclic triterpenes and pentacyclic triterpenes, among which the former mainly include lanostane, euphorane, dammarane, and cucurbitane, as well as protostance, meliacane, cycloartane, etc. (Figure 1A); and the later mainly include oleanane, ursane, lupinane, friedelane, fernane, isofernane, hopane, and isohopane (Figure 1B), and so on [11,12,13]. The steroidal saponins are formed by the condensation of steroidal sapogenins and glycosyl groups. It consists of 27 carbon atoms, and their basic carbon skeleton is a derivative of spirostane. According to the configuration of C_25_ in the spirostane structure and the cyclization state of the F ring, they are further divided into four types: spirostanol, isospirostanol, furostanol, and pseudospirostanol (Figure 1C) [14].

Triterpene saponins mostly consist of carboxyl groups and are acidic, so they are often called acidic saponins; while steroidal saponins, on the contrary, are neutral saponins. There also exist some differences in the division of triterpene saponins and steroidal saponins in nature. The former is mainly derived from dicotyledonous plants, and the common branches are in *Leguminosae*, *Araliaceae*, *Campanulaceae*, *Cucurbitaceae*, *Ranunculaceae*, *Umbelliferae*, and *Rhamnaceae*, etc.; the latter is mainly obtained from monocotyledonous plants, such as *Liliaceae*, *Dioscoreaceae*, *Solanaceae*, *Amaryllidaceae*, *Agaveaceae*, *Scrophulariaceae*, and *Rhamnaceae*, etc. [15,16].

Saponins possess a wide range of biological activities, such as anti-inflammatory, antitumor, antibacterial, antiviral, immune regulation, analgesic, neuroprotective, hepatoprotective, antihyperlipidemic, hypocholesterolemic, hypotensive, and so on [17,18,19,20]. Next, we will focus on their effects and mechanisms in the treatment of neuropathic pain (Table 1).

## 3. Preclinical Evidence for Saponins in Neuropathic Pain

### 3.1. Ginsenosides

Ginsenosides are the major biologically active components of *Ginseng*, which have a wide range of pharmacological activities. According to the skeleton of their aglycones, ginsenosides can be classified into two groups, tetracyclic triterpene dammarane-type saponins (protopanaxadiol (PPD)-, protopanaxatriol (PPT)-type) (Figure 2A) and tetracyclic triterpene oleanane-type saponins [48,49,50]. So far, more than 100 different ginsenoside monomers have been isolated, such as ginsenosides Rb1, Rb2, Rc, Rd, Re, Rg1, and Rf, the pharmacological and pharmacokinetic properties of which are different [51,52].

Data based on animal models have shown that ginsenosides play a beneficial role in neuropathic pain. In a study conducted by Jee Youn Lee et al. [21], peripheral and central neuropathic pain was induced by tail nerve injury or contusive spinal cord injury (SCI) in male SD rats, respectively. Remarkable analgesic effects were shown after the application of oral total saponin extract (TSE), ginsenoside Rb1. The research found that TSE and ginsenoside Rb1 inhibited the activation of microglia/astrocytes, and attenuated inflammatory factors levels, such as interleukin-1β (IL-1β), interleukin-6 (IL-6), inducible nitric oxide synthase (iNOS), and cyclooxygenase-2 (COX-2). Further results demonstrated that TSE, ginsenoside Rb1, and Rb1-metabolite-compound, K, also exerted analgesic effects that might be mediated through the estrogen receptor. Other research conducted by Gao Chao et al. reported that intrathecal injection of ginsenoside Rg1 significantly inhibited chronic constrictive injury (CCI)-induced thermal hyperalgesia in a dose-dependent manner. It might be mediated by inhibiting the expression of phosphorylated p38 mitogenactivated protein kinase (p-p38MAPK) and nuclear factor kappa-B (NF-κB) subunit phosphorylated p65, and the activation of ionized calcium binding adaptor molecule-1 (IBA-1) in the spinal microglia, resulting in downregulation of the central sensitization [22]. In addition, other studies showed that ginsenoside Rf robustly decreased IL-1β and IL-6, but increased the expression of IL-10 in the dorsal root ganglion (DRG), in both the spinal cord and DRG of CCI rats [23]. Thus, ginsenoside Rf may adjust the balance between proinflammatory and anti-inflammatory factors to promote its antinociceptive effect in neuropathic pain.

Many studies have revealed the key role of proinflammatory cytokines in the pathophysiology of neuropathic pain [53,54,55,56]. The above studies all explained that related ginsenosides inhibited inflammation through different pathways to relieve neuropathic pain. Furthermore, other studies have shown that ginsenoside Rb1 inhibits neuronal apoptosis [24] and promotes the neurogenesis and regulates the expressions of brain-derived neurotrophic factor (BDNF) and caspase-3 to play a neuroprotective effect [57]. On the other hand, clinically chronic pain patients are often accompanied by depression, and some depressive patients also have chronic somatic pain symptoms [58]. Therefore, the relationship between pain and the occurrence of depression has become the focus of recent studies. It has shown that intraperitoneal injection of ginsenoside Rg2 not only alleviates the mechanical allodynia and thermal hyperalgesia, but also relieves anxiety and depression in CCI rats [59], though its underlying mechanism needs to be further explored. So far, most of the analgesic mechanisms of ginsenosides in the neuropathic pain are limited to the exploration of inflammatory factors, lacking in-depth analysis of its targeted molecular targets. In addition, whether the regulatory effects of ginsenosides are related with different neuropathic pain-related brain regions is still largely unknown. Further studies focusing on these points may provide a research basis for the precise regulation of drugs.

### 3.2. Saikosaponins

Saikosaponins are derived from *Bupleurum* or *Bupleurum* scorzonerifolium in the *Umbelliferae*, one of the traditional Chinese herbal medicines, and are the main active ingredients of *Bupleurum* [60]. So far, more than 100 kinds of saikosaponins have been isolated from *Bupleurum*, the main ones of which are oleanane and ursolic pentacyclic triterpene saponins [61,62,63,64,65]. According to their chemical structure, saikosaponins are divided into -A, -B, -C, -D, -M, -N, -P, and -T categories, and Saikosaponin D (SSD) is considered to be the most active one, followed by Saikosaponin A (SSA) [66,67]. Their chemical structures are shown in Figure 2B.

Both in vivo and in vitro experimental studies have shown that saikosaponins can inhibit the activation of transient receptor potential ankyrin 1 (TRPA1) and significantly reduce the nociceptive response of animals induced by allyl isothiocyanate (AITC) [25]. Molecular docking and site-directed mutagenesis analyses demonstrated that saikosaponins bind to the TRPA1 hydrophobic pocket near the Asn855 residue, which once mutated to Ser and was previously united with enhanced pain perception in humans [25,68]. Gyeongbeen also reported that multiple administrations of SSD could significantly relieve mechanical hypersensitivity induced by vincristine, which was carried out partially by suppressing the activity of TRPA1 [25]. Therefore, it can be further speculated that SSD might play a certain therapeutic role in the neuropathic pain that is induced by chemotherapeutics, diabetes, or CCI, in which the expression and sensitivity of TRPA1 were changed as well, resulting in abnormal pain response and perception [69,70,71,72,73]. However, the analgesic effect of SSD is different between streptozotocin (STZ)- and paclitaxel-induced pain models. Short-term oral administration was effective in the former, while multiple administrations were required for the pain relief of the latter [26]. This indicates that the analgesic effect of SSD may not only act as an antagonist of TRPA1, but also exert anti-inflammatory activity to reduce the oxidative stress caused by nerve damage. Related research reported that SSD could restrain the translocation of the glucocorticoid receptor to the mitochondria, and decrease the H_2_O_2_-induced phosphorylation of extracellular-regulated kinase (ERK), c-Jun N-terminal kinase (c-JNK), and p38MAPK to downregulate the activity of neuronal PC12 cells [27,74,75].

It is well known that activation of NF-κB in both DRG and spinal cord neurons is associated with the transduction and processing of nociceptive messages. Therefore, inhibition of NF-κB can alleviate chronic painful states [76]. Studies have shown that SSA alleviates neuropathic pain by inhibiting CCI-induced elevation of p-p38 MAPK and NF-κB levels in the spinal cord [28]. In addition, cytokine dysregulation is one of the characteristic manifestations of neuropathic pain symptoms [77]. It could also be found that SSA significantly inhibited the expression of certain immune-related cytotoxic factors, including COX-2 and iNOS, and, likewise, the pro-inflammatory cytokines, such as TNF-α, IL-1β, and IL-6. Meanwhile, the expression of the important anti-inflammatory cytokine IL-10 was significantly upregulated, suggesting that it had anti-inflammatory activity in lipopolysaccharide (LPS)-stimulated macrophages [29,30]. Further research showed that SSA blocked the NF-κB signaling pathway by preventing phosphorylation of the NF-κB inhibitor α (IκBα), thereby allowing p65 NF-κB to remain in the cytoplasm, preventing it from translocating to the nucleus. In addition, SSA inhibited the MAPK signaling pathway by downregulating the phosphorylation of p38 MAPK, c-JNK, and ERK to exert the anti-inflammatory activity [30]. On the basis, SSA appeared to counteract the neurological function deficits after traumatic brain injury via inbiting aquaporin-4 (AQP-4) and matrix metalloprotein-9 (MMP-9) to account for its neuroprotective effects [31]. On the other hand, a study of Seong Shoon Yoon et al. expressed that SSA exhibited a significant inhibitory effect on morphine-reinforced behavior and drug addiction predominantly via mediating GABAB receptors [78,79]. Davoud Ahmadimoghaddam et al. reported that *Bupleurum falcatum* L. roots essential oil, of which SSA was one of the main constituents [32], exerted its antinociceptive and antiallodynic effects through the regulation of L-arginine-NO-cGMP-KATP channel pathways, as well as interaction with opioid, peroxisome proliferator-activated, and cannabinoid receptors [32]. The voltage-gated sodium channel Nav1.7 is a tetrodotoxin-sensitive sodium channel subtype and is encoded by SCN9A. It is well known that the dysfunction of Nav1.7 has the correlation with pain disorders [80]. Relevant research showed that SSA displayed the analgesic effects on the thermal pain and formalin-induced pain in mice via strong inhibitory effect on the peak currents of Nav1.7 [33].

The above studies have shown that SSD and SSA can exert analgesic effects in different neuropathic pain models through multiple pathways, and their mechanisms of action have similarities and differences. In the follow-up, we can combine their structural characteristics with the mechanisms of action for deep analysis to provide a research basis for the precise regulation of the targets.

### 3.3. Astragalosides

*Astragali* Radix, the dried roots of *Astragalus membranaceus* (Fisch.) Bge. *var. mongholicus* (Bge.) Hsiao, or *Astragalus membranaceus* (Fisch.) Bge., is known as a high-grade traditional Chinese medicine [81]. There are three main types of compounds in astragalus: saponins, flavonoids, and polysaccharides, and triterpene saponins are the major constituents [82,83,84]. It is reported that more than 40 kinds of saponins have been isolated and identified from the dried astragalus roots via HPLC and GC-MS, such as astragalosides I–VIII, acetylastragaloside, isoastragaloside I, III, astramembrannin II, cycloastragenol, cycloascauloside B, brachyoside B, astrasieversianin X, etc. [85,86,87,88]. Among these, astragaloside IV (AS-IV) is known as the major active ingredient and qualitative control biomarker. AS-IV is 3-*O*-beta-d-xylopyranosyl-6-*O*-beta-d-glucopyranosyl-cycloastragenol (Figure 2C), the molecular formula is C_41_H_68_O_14_ [89].

It is generally accepted that the transient receptor potential vanilloid 1 (TRPV1) channel is a polymodal receptor for various stimuli such as noxious heat and capsaicin, and is also an important pain sensor [90,91]. TRPV1 is overexpressed in Aδ fibers and C fibers in the situation of inflammation or nerve injury [92]. In addition, purinergic P2 × 3 receptors are ligand-gated nonselective cation channels, highly selectively expressed in small-diameter and medium-diameter sensory neurons related to nociceptive information, and play a key role in the generation and maintenance of pathological pain [93,94]. In the research by Guo-Bing Shi et al., AS-IV not only dramatically downregulated the expression of TRPV1 in Aδ fibers to remarkably upregulate the nociceptive threshold, but also inhibited P2 × 3 expression in DRG neurons to attenuate the mechanical allodynia [34]. Meanwhile, AS-IV restored the histological structure of the damaged sciatic nerve by accumulating glial cell-derived neurotrophic factor family receptorα1 (GFRα1), the glial cell derived neurotrophic factor (GDNF) selective receptor, in the debris of myelin between the Schwann cells and the damaged axon [34]. It also reduced the levels of GFRα1 and GDNF in DRG, which were highly expressed and induced by CCI, contributing to the restoration of injured nerve fibers [95,96].

In the peripheral nervous system, the appropriate dose of AS-IV could also greatly promote the regeneration of peripheral nerves [35]. Growth-associated protein 43 is lower in spinal cord segments L4–6 but active in growing neuronal axons in normal Balb/c mice. As a particular biomarker in nerve injury, it plays a vital role in nerve growth, and strongly associates with neuronal axon growth [36,89,97,98]. Previous research showed that AS-IV significantly upregulated the expression of growth-related protein 43 in regenerated nerve tissue, thereby increasing the number and diameter of myelinated nerve fibers in the sciatic nerve of mice, while elevating motor nerve conduction velocity and action potential amplitude [36]. Moreover, AS-IV also conducted analgesic effects on peripheral neuropathy in STZ-induced diabetic rats. Firstly, it reduced blood glucose and glycosylated hemoglobin (HbA_1_C) levels, and increased plasma insulin levels in diabetic rats [37]. It is crucial to control the levels of HbA_1_C because its concentration is closely related to the incidence of diabetes-related complications, which has been proven by clinical trials [99]. Secondly, AS-IV enhanced the activity of glutathione peroxidase in nerves, suppressed the activation of aldose reductase in erythrocytes, and decreased the accumulation of advanced glycation end products in both nerves and erythrocytes, which might not only activate the cellular antioxidant defense system, but also aggrandize the ability of antioxidative stress injury on peripheral nerves. Thirdly, AS-IV acted as the AR inhibitor, and then enhanced Na^+^, K^+^-ATPase activity, improved the delayed motor nerve conduction velocity, increased nerve blood flow, and prevented structural nerve fiber damage to correct peripheral nerve defects [37].

### 3.4. Diosgenin

Diosgenin is a naturally occurring steroidal sapogenin and is abundant in nature. Primary sources of diosgenin include the three *Dioscorea* species and one *Heterosmilax* species, namely, *D. zingiberensis*, *D. septemloba*, *D. collettii*, and *H. yunnanensis* [100]. Diosgenin can also be obtained from fenugreek (*T. foenum graecum Linn*) and *Costus speciosus* [101,102,103]. It is a C27 spiroketal steroid sapogenin, 3β-hydroxy-5-spirostene (Figure 2D), and its molecular formula is C_27_H_42_O_3_ [104]. As a representational phytosteroid, diosgenin is an important basic raw material for the production of steroid hormone drugs and has received increasing attention in the pharmaceutical industry for decades [105]. In addition, diosgenin itself has a wide range of biological effects. The following studies mainly describe its role in neuropathic pain.

Neuropathic pain, one of the common complications of diabetes mellitus, manifests as increased sensitivity to noxious stimuli [106]. To evaluate the effects of diosgenin in the treatment of diabetes-induced neuropathic pain, an in vivo study was performed on a rat model of STZ-induced diabetes. It was demonstrated that diosgenin upturned mechanical and thermal nociceptive thresholds and lowered pain scores at the late phase of the formalin test in diabetic rats [38]. Since elevated oxidative stress is one of the key factors in diabetes-related neurological dysfunction, it can lead to vascular dysfunction, resulting in intraneural hypoxia, which can lead to impaired motor and sensory nerve function [107,108]. Studies showed that diosgenin could reduce the content of malondialdehyde (MDA) in serum, DRG, and sciatic nerve of diabetic rats and restored the activities of superoxide dismutase (SOD) and catalase, thereby inhibiting oxidative stress and enhancing the function of the antioxidant defense system [38]. Furthermore, NF-κB, an important nuclear transcription factor, is responsible for the control of genes encoding inflammation and nociception-related mediators [109]. Upregulation of NF-κB in the DRG neurons of diabetic rats has been proven, and its inhibition significantly reduces nociceptive responses [110,111,112]. It reported that diosgenin downregulated the NF-κB p65/p50 signaling pathway in the LPS-induced lung injury model [113]. However, based on the available reports, there is no specific experimental research regarding whether diosgenin exerts its analgesic effect in diabetes-induced neuropathic pain by regulating NF-κB, and related research needs to be further developed. Nerve growth factor (NGF), as a neurotrophic factor, is a protein factor that plays a vital role in the maintenance of the growth, development, and function of sympathetic and sensory neurons. It stimulates the axon growth, maintains the axon size, prevents the postinjury death of mature neurons, and regulates various functions of the nervous system, including synaptic plasticity and neurotransmission [114,115]. In diabetic neuropathy, the function of NGF is impaired and the expression of NGF-related genes is modified, which are important factors in the progress of diabetic neuropathic pain. A study conducted by Tong Ho KANG et al. revealed that diosgenin upregulated the level of NGF in the sciatic nerve of diabetic rats. The comparable effects also reported that diosgenin increased the neurite outgrowth of PC12 cells, enhanced the sciatic nerve conduction velocity of diabetic mice by inducing NGF, reduced myelin disturbance, increased the area of myelinated axons, and improved the signal transmission of damaged axons, thereby alleviating diabetic neuropathic pain [39].

In addition to the diabetic neuropathy model, the role of diosgenin in the treatment of neuropathic pain has also been reported in the CCI rat model. In 2017, Wei-Xin Zhao et al. performed an in vivo study, and the results demonstrated that diosgenin could upregulate CCI-reduced mechanical withdrawal threshold and thermal withdrawal latency. This was due to the fact that diosgenin not only inhibited CCI-induced elevation of proinflammatory cytokines TNF-α, IL-1β and IL-2, but also suppressed oxidative stress in the spinal cord. Moreover, diosgenin remarkably restrained the expression of p-p38 MAPK and NF-κB in the spinal cord and eased neuropathic pain in CCI rats by inhibiting the activation of p38 MAPK and NF-κB signaling pathways [40]. In other research [41], sciatic-crushed-nerve injury in rats decreased the sciatic function index, which was widely used to evaluate functional gait [116], increased the c-Fos expression in the ventrolateral periaqueductal gray and paraventricular nucleus, restrained recovery of locomotor function caused by the overexpression of BDNF, and aggrandized expressions of COX-2 and iNOS that responded to inflammation. Fortunately, diosgenin was able to significantly improve the above pathological states, and exploited potential abilities in pain control and functional recovery after peripheral nerve injury.

### 3.5. Saponin-Rich Extracts of O. sanctum

In addition to the analgesic effects of the above four plant saponins that have been identified with clear structures, saponin-rich extracts of *O. sanctum* have also been found with similar effects. *O. sanctum* is the aerial part of Ocimum basilicum, a plant of the Labiatae family. Modern pharmacological studies have illustrated that the chemical composition of *O. sanctum* is complex and the types are diverse, including volatile oils, flavonoids and their glycosides, coumarins, phenylpropanoids, and fatty acids, mainly volatile oils and flavonoids and their glycosides [117]. In addition, a variety of saponins have been isolated from the alcoholic extract of *O. sanctum* [118], the most important of which are pentacyclic triterpenoid saponins that are dominated by ursolic and oleanolic acids [119,120,121], and have a wide range of pharmacological effects.

Oxidative stress [122] and alterations in calcium homeostasis [123] are thought to be closely associated with neuropathic pain. During neurological disorders, dysfunction of the intracellular calcium regulatory system produces oxidative stress [124], and increases in free radicals lead to neuronal degeneration and apoptosis. On the other hand, metabolic abnormalities [125], formation of protein aggregates [126], and changes in membrane permeability [127] caused by oxidative stress all increase calcium levels, and they act together to promote the deterioration of neuropathic pain. *O. sanctum* has a good antioxidant effect [128,129], protects against free radical damage [130], and is able to reduce calcium levels [42]. *O. sanctum* is used as a neurotonic in parts of India for the relief of headache, joint pain, and muscle pain. In the experiments conducted by Muthuraman et al., the administration of *O. sanctum* attenuated sciatic nerve transection-induced peripheral neuropathy and motor in-co-ordination, attenuated the amputation-induced reduction in thiobarbituric acid reactive species, total calcium, and glutathione levels in a dose-dependent manner [42]. It suggested that the analgesic effect of *O. sanctum* might be related to its antioxidation and reduction of calcium levels. Additionally, in other studies, treatment with *O. sanctum* and its saponin-rich fraction reduced neuropathic pain caused by chronic constrictive injury and chemotherapeutic agent vincristine, associated with its effects on the oxidative stress and calcium levels [42,43]. Based on the above findings, it can be observed that the downregulation of calcium levels by *O. sanctum* administration may be due to a direct effect on or secondary to a decrease in oxidative stress. Then, it produces an antinociceptive or antiapoptotic effect on neurons. It has been reported that Saponins have antioxidant [131] and calcium lowering effects [132]. Thus, the antinociceptive effect of *O. sanctum* saponins may be constructed through direct or indirect reduction of calcium levels.

There is also evidence that *O. sanctum* leaves and seeds reduce uric acid levels in rabbits [133], and elevated uric acid levels are associated with gouty arthritis and other joint inflammation [134]. The ethanolic extract of *O. sanctum* can be antinociceptive, and involves the interaction of neurotransmitter systems such as opioid receptors and norepinephrine [135]. These studies support the traditional use of *O. sanctum* for the treatment of inflammation and pain, without excluding the effects of other active ingredients such as flavonoids and phenols.

## 4. Conclusions and Perspective

In this review, we summarized the role of five well-studied and representative saponins in reducing neuropathic pain with the following key points: (1) Saponins can effectively reduce neuropathic pain in different nerve injury models, such as the spared nerve injury, spinal nerve ligation, partial sciatic nerve injury, diabetes-induced neuropathy, chemotherapy-induced neuropathy, and CCI models. (2) The analgesic effects of saponins are mainly related to their anti-inflammatory, immunol regulatory, antioxidative stress and neuroprotective activities.

Of course, further prospective research is also needed to address the following problems. Luckily, with the development of advanced technology, we can combine these technical methods to promote the further development of related research listed below (Figure 3):

(1) Saponins are widely distributed in nature. In addition to the above five saponins, there are some other less studied saponins (saponins from *Tribulus terrestris* [44], escin [45,46], etc.) that have been found to have therapeutic effects in animal models of neuropathic pain. Therefore, according to the species characteristics of plants, more saponin compounds can be explored to provide a material basis for new drug research. Facing more and more different types and structures of saponins, knowing how to efficiently and quickly screen out the effective parts is an urgent problem to be solved. In addition, the extraction rate of some saponins from plants is not high enough, so it is possible to synthesize saponins based on their chemical structural characteristics.

In order to quickly and efficiently screen out biologically active saponins, high content screening technology can be used. It can realize multitarget, multiparameter detection of saponins. Moreover, combined with computer-aided drug design technology, the structure optimization and target docking of saponin derivatives can be achieved, which provides a theoretical basis for the synthesis of saponin derivatives with an efficient analgesic effect.

(2) At present, the research on the mechanism of action of saponins extracted from traditional Chinese medicine in neuropathic pain is relatively scattered, relatively independent, and the research depth is not enough. In the follow-up, the relationship between the structural properties of saponins and their targets need to be analyzed, and the effective sites of action may be synthesized and modified to provide a research basis for multitarget therapy. In-depth analysis of the pathogenesis of neuropathic pain, a full understanding of its circuit abnormalities, and an insight into the circuit basis and function characteristics of saponins to relieve pain can provide directions for the subsequent development of targets. Combined with optogenetics, pharmacogenetics, and other regulatory methods, deep and systematic research for this purpose can be well conducted.

(3) The ultimate goal of drug discovery is to treat diseases, so clinical research is of great importance. Some analgesics currently in clinical use have the problems of low safety, large side effects, and unsatisfactory analgesic effects. The current literature supporting saponins for the treatment of neuropathic pain are limited to animal models, and future studies are needed to evaluate the efficacy of these saponins in the clinic.

In view of the shortcomings of current clinical drugs, it is possible to consider the study of the combination of existing drugs and traditional Chinese medicines containing saponins or extracted saponins to offer more alternatives for the treatment of neuropathic pain. Research has demonstrated that ginsenoside Rf potentiates U50-induced analgesia and inhibits tolerance to its analgesia via nonopioid, non-dihydropyridine-sensitive Ca^2+^, and non-benzodiazepine-GABAergic mechanisms in mice [47]. Meanwhile, combination therapy can reverse drug resistance to a certain extent. In addition, electroacupuncture treatment of neuropathic pain and pain-induced negative emotions are gradually extensive [136,137]. With the improvement of new materials and pharmaceutical preparation technology, the technology of electroacupuncture with drug-loading capabilities, such as saponins, has gradually become possible. The synergistic and targeted delivery of acupuncture and medicine provides a new prospect for the treatment of neuropathic pain.

Overall, even with the aforementioned obstacles and problems, saponins remain the valuable drug candidates for the treatment of neuropathic pain. It is believed that with the provision of future research technologies and in-depth research, the reliability of saponins against neuropathic pain will be greatly improved, which will promote their application in actual neuropathic pain treatment.

## 5. Strengths and Limitations of the Review

The strength of this review is its systematical and comprehensive summary of the effects and mechanisms of different saponins on neuropathic pain. At the same time, according to the current research technologies and progress, some limitations of the previous research are pointed out and some constructive opinions are put forward.

On the other hand, this paper also has a limitation. This review is based on the structure and type of saponins, which leads to the inability to systematically summarize the role of saponins in different types of neuropathic pain, which are only introduced separately in each part.

## Figures and Tables

**Figure 1 molecules-27-03956-f001:**
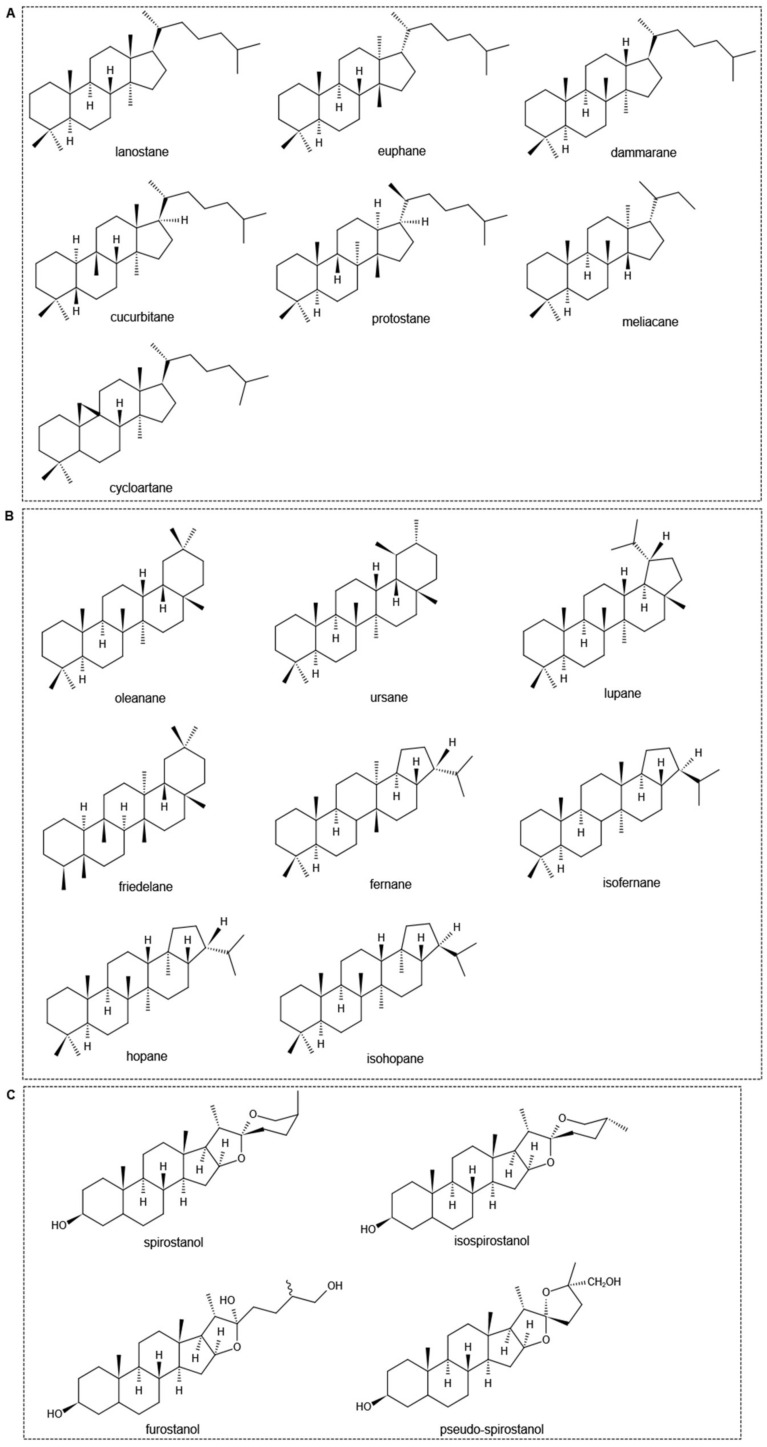
Chemical structures of different subgroups of tetracyclic triterpenoid saponins (**A**), pentacyclic triterpenoid saponins (**B**), and steroidal saponins (**C**).

**Figure 2 molecules-27-03956-f002:**
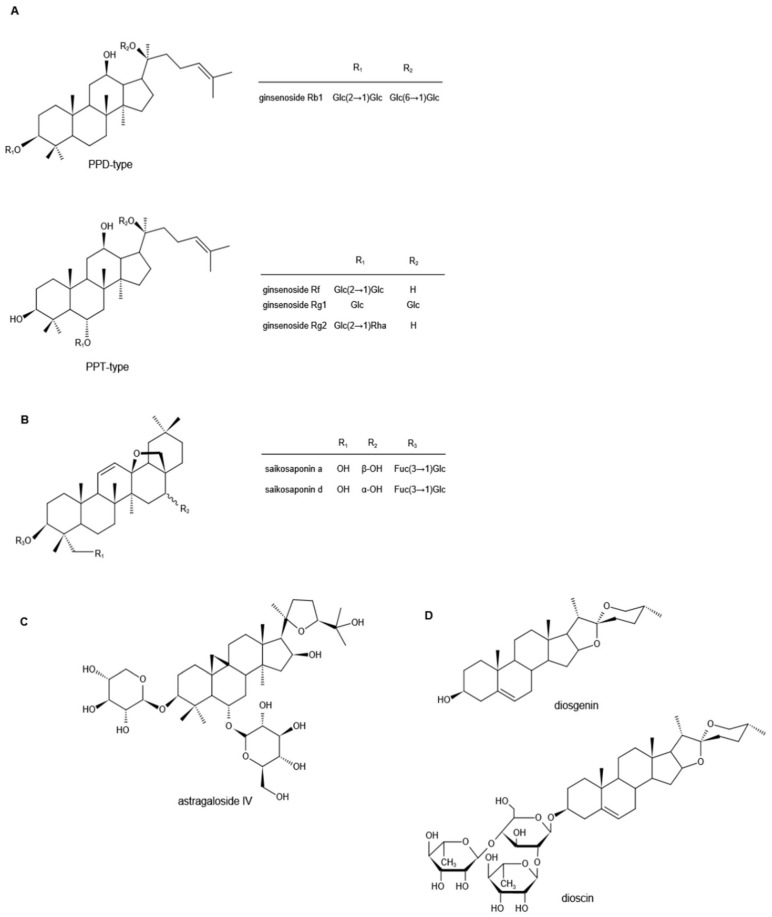
Chemical structures of ginsenosides (**A**), saikosaponins (**B**), astragaloside IV (**C**), and diosgenin and dioscin (**D**) (Note: PPD, protopanaxadiol; PPT, protopanaxatriol; Glc, glucoside; Rha, rhamnoside; Fuc, fructoside).

**Figure 3 molecules-27-03956-f003:**
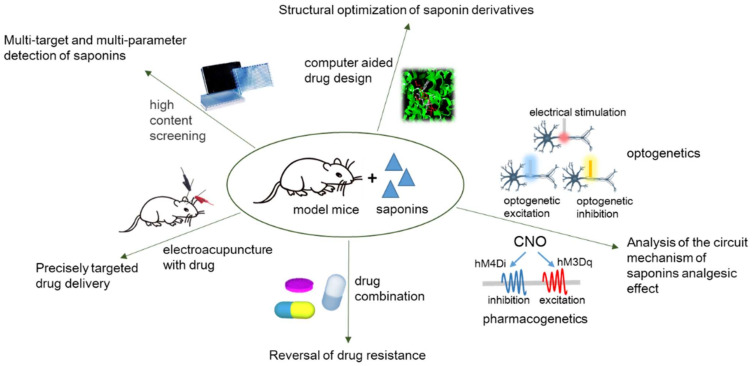
Schematic diagram of the direction of in-depth analysis of saponins in the treatment of neuropathic pain in future research.

**Table 1 molecules-27-03956-t001:** Effects of saikosaponins on neuropathic pain.

Saponins	Animals/Cells	Dose mg/kg	Effects/Behavioral Evaluation	Mechanisms of Action	Reference
total saponin extract (TSE), ginsenoside Rb1, Rb1 metabolite compound K	male Sprague–Dawley rats	TSE 50 mg/kg, Rb1 12.5 mg/kg, compound K 7 mg/kg (p.o.)	↓TNI-induced mechanical, cold, and warm allodynia; ↓SCI-induced mechanical allodynia and thermal hyperalgesia; →Basso–Beattie–Bresnahan locomotor scale	↓IL-1β, IL-6, iNOS, COX-2; ↓microglialand astrocyte activation	Jee Youn Lee [21]
ginsenoside Rg1	male Sprague–Dawley rats	2, 5, 10 ug/uL (intrathecal injection); 4 days	↓CCI-induced thermal hyperalgesia;	↓IBA-1, OX-42; ↓microglialover-activation; ↓p38MAPK/NF-κB signaling pathway	Gao Chao [22]
ginsenoside Rf	male Sprague–Dawley rats	0.5, 1.5, 3 mg/kg (i.p.); 1 day, 7 days, 14 days, 21 days	↓CCI-induced mechanical hyperalgesia; ↓CCI-induced immobility in the forced swimming test	↓IL-1β and IL-6 in both the spinal cord and the DRG; ↑IL-10 in the DRG but not in the spinal cord	Yangyi Li [23]
ginsenoside Rb1	male and female Sprague-Dawley rats	10 mg/kg (i.p.); 1–7 days	↑SCI-induced reduction of Basso–Beattie–Bresnahan locomotor scale	↓neuronal damage; ↓apoptotic rate in spinal cord neurons. ↑ AQP4 expression	Fei Huang [24]
saikosaponin A; saikosaponin B1; saikosaponin B2; saikosaponin C; saikosaponin D; saikosaponin F; B. falcatum extract	male ICR mice/HEK293 cells	B. falcatum extract 50 mg/kg, saikosaponin D 20 mg/kg (p.o.)	↓AITC-induced nociceptive behaviors; ↓vincristine-induced mechanical hypersensitivity	saikosaponins are TRPA1 antagonists	Gyeongbeen Lee [25]
saikosaponin D	male ICR mice	10 mg/kg (p.o.), 1 day and 15 days	↓ STZ-induced mechanical hypersensitivity and paclitaxel-induced mechanical allodynia	____	Gyeongbeen Lee [26]
saikosaponin D	PC12 cells	200, 300, 400 μg/mL	↓H_2_O_2_-induced decrease in cell viability; ↓apoptosis rate; ↓caspase-3 activation and poly-ADP-ribose polymerase cleavage; improved the nuclear morphology	↓H_2_O_2_-induced release of malonic dialdehyde MDA and lactate dehy-drogenase; ↑SOD; ↓apoptotic rate; ↓ H_2_O_2_-induced p-ERK, p-c-JNK, p-p38MAPK	Xuemei Lin [27]
saikosaponin A	male Sprague–Dawley rats	6.25, 12.50, 25.00 mg/kg (i.p.), 14 days	↓CCI-induced mechanical allodynia and thermal hyperalgesia	↓TNF-α, IL-1β, IL-2 in spinal cord; ↓p38MAPK/NF-κB signaling pathway	Xin Zhou [28]
saikosaponin A; saikosaponin D	male BALB/c mice; male Sprague–Dawley rats; Raw264.7 cells	20, 10, 5 mg/kg (p.o.);	↓carrageenan-induced rat paw edema; ↓acetic acid-induced evans blue dye leakage	↓NO, PGE2, IL-6, TNF-α, iNOS, COX-2 in LPS-induced RAW264.7 cells; ↓NF-κB signaling pathway	Chun-Ni Lu [29]
saikosaponin A	Raw264.7 cells	3.125, 6.25, 12.5, 25 μM	____	↓IL-1β, IL-6 TNF-α, iNOS, COX-2 in LPS-induced RAW264.7 cells; ↓MAPK/NF-κB signaling pathway	Jie Zhu [30]
saikosaponin A	male Sprague–Dawley rats	20 mg/kg (i.v.); 3 days	↑neurological functions andcognition; ↓brain edema and blood brain barrier permeability after controlled cortical impact	↓AQP-4, MMP-9, MAPK, c-JNK, TNF-α, IL-6; ↓MAPK signaling pathway	Xiang Mao [31]
*Bupleurum falcatum* L. roots essential oil (BFEO); Saikosaponin A	male Swiss mice	BFEO 25, 50, 100 mg/kg (p.o.); SA 6, 12, 25 mg/kg (p.o.)	↑the antinociceptive activity in formalin-induced paw licking test, ↓mechanical allodynia, →locomotor action	↑the L-arginine–NO–cGMP-KATP channel pathway	Davoud Ahmadimoghaddam [32]
saikosaponin A	male Kunming mice; Nav1.7-CHO cells	2.5, 5, 10 mg/kg (i.g); 100 nM;	↓thermal pain and formalin-induced nociceptive responses	inhibitory effect on Nav1.7	Yijia Xu [33]
astragaloside IV	male Sprague–Dawley rats	15, 30, 60 mg/kg (i.p.), 23 days	↓CCI-induced mechanical allodynia and thermal hyperalgesia; ↑CCI-induced reduction of nerve conduction velocity; →locomotor action	↓P2 × 3, TRPA1 and TRPV1 in the DRG; restoring the histological structure of the damaged sciatic nerve by accumulating GFRα1	Guo-Bing Shi [34]
astragaloside IV	Sprague–Dawley rats	0, 50, 100, 200μM	↑regeneration rate across the wide gap; ↑myelinated axons; ↑evoked action potential; ↓nerve regeneration	plays a dual role in anastomosis	Chun-Yuan Cheng [35]
astragaloside IV	BALB/c mice	2.5, 5, 10 mg/kg (i.p.)	↑denervating the left sciatic nerve-induced the number and diameter of myelinated nerve fibers; ↑motor nerve conduction velocity and action potential amplitude in the sciatic nerve	↑growth-associated protein-43 expression; ↑pheral nerve regeneration and functional reconstruction	Xiaohong Zhang [36]
astragaloside IV	male Sprague–Dawley rats	3, 6, 12 mg/kg (p.o.), 12 days	↑pain threshold in STZ-diabetic rats; ↑motor nerve conduction velocity	↓blood glucose concentration and HbA1C levels; ↑plasma insulin levels, the activity of glutathione peroxidase in nerves; ↓ the activation of aldose reductase in erythrocytes and advanced glycation end products; ↑Na+,K+-ATPase activity	Junxian Yu [37]
diosgenin	male albino Wistar rats	40 mg/kg (i.g), 35 days	↓mechanical hyperalgesia and thermal hyperalgesia and pain score in STZ-diabetic rats;	↓MDA, ↑SOD and catalase activity; ↓NF-κB and IL-1β	Zahra Kiasalari [38]
diosgenin	male ICR mice, male Sprague–Dawley rats; PC12 cells, C6 glioma cells	10 mg/kg (p.o.); 0.1–10 mg/mL	↑NGF levels in alloxan-diabetic rats; ↑nerveconduction velocities	reverses functional and ultrastructural changes and induces neural regeneration	Tong Ho Kang [39]
diosgenin	male Sprague–Dawley rats	10, 20, 40 mg/kg (i.p.), 14 days	↓CCI-induced mechanical allodynia and thermal hyperalgesia.	↓TNF-α, IL-1β, IL-2, and oxidative stress; ↓p38MAPK/NF-κB signaling pathway	Wei-Xin Zhao [40]
diosgenin	male Sprague–Dawley rats	25, 50, 100 mg/kg (p.o.), 7 days	↑functional locomotor recovery following sciatic crushed nerve injury	↓nerve injury-induced increase in BDNF, TrkB, COX-2, and iNOS expressions	Byung-Ki Lee [41]
ocimum sanctum, saponin-rich extracts	Wistar albino rats	100 and 200 mg/kg (p.o.), 14 days	↓CCI-induced cold-allodynia, heat-hyperalgesia, mechanical hyperalgesia and tail cold-hyperalgesia	↓oxidative stress and calcium levels	Gurpreet Kaur [42]
ocimum sanctum, saponin-rich extracts	Wistar albino rats	100 and 200 mg/kg (p.o.), 14 days	↓ vincristine-induced cold-allodynia, heat-hyperalgesia, mechanical hyperalgesia and tail cold-hyperalgesia	↓oxidative stress and calcium levels	Gurpreet Kaur [43]
saponins of Tribulus terrestris	Wistar rats of either sex	25, 50, 100 mg/kg (p.o.)	↓vincristine-induced mechanical hyperalgesia and allodynia; ↓chemical-induced nociception	↓TNF-α, IL-1β, and IL-6; ↑ nerve conduction velocity, neurotransmitters, l-glutamic acid and l-aspartic acid	Mrinmoy Gautam [44]
escin	male Kunming mice, male Sprague–Dawley rats; PC12 cells	7, 14, 28 mg/kg (i.g.), 14 days; 15, 25, 35 mg/kg (i.g.), 3 days; 2.5, 5, 10 μM	↓CCI-induced thermal hyperalgesia; ↓ formalin-induced nociceptive responses	↓TLR-4/NF-κB signal pathway; ↓GFAP, NGF	Liudai Zhang [45]
escin	male Sprague–Dawley rats	4 mg/kg (i.p.), 7 days	↓Paclitaxel-induced mechanical allodynia and thermal hyperalgesia	↑LC3II expression, ↓p62expression levels	Yan Fang [46]
ginsenoside Rf	Swiss male mice	10^−14^, 10^−12^, and 10^−10^ mg/kg, (i.p.), 6 days; 10^−12^–10^−2^ mg/mL	↑U50-induced analgesia, ↓tolerance	nonopioid and non-dihydropyridine-sensitive Ca^2+^ channel mechanisms; non-benzodiazepine-GABAAergic mechanisms	Kumar V.S. Nemmani [47]

Note: ↑ increase/enhanced/upregulate; ↓ decrease/inhibit/prevent/attenuate/downregulate; TSE, total saponin extract; TNI, tail nerve injury; SCI, spinal cord injury; CCI, chronic constrictive injury; DRG, the dorsal root ganglion; AITC, allyl isothiocyanate; STZ, streptozotocin; LPS, lipopolysaccharide; BFEO, *Bupleurum falcatum* L. roots essential oil.

## Data Availability

Not applicable.

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
