# Peer review of "The Role of Saponins in the Treatment of Neuropathic Pain"

_molecules, 2022, doi:10.3390/molecules27123956_

Round 1

Reviewer 1 Report

Exellent manuscript with a new idea, important for the daily practice in chronic pain patients, clearly structured

Author Response

Reply: Thank you very much for your time to review our work and your valuable comments.

Reviewer 2 Report

The paper of Tang et al., entitled ”The role of saponins in the treatment of neuropathic pain” is a very interesting and well documented paper that can be published in Molecules journal, after minor correction of English language.

Author Response

Reply: Thank you very much for your valuable comments. The English of manuscript has been re-checked by an English-speaking native. All the revisions are highlighted by red color in the revised manuscript.

Reviewer 3 Report

A interesting and well-written article. I suggest, as minor corrections, to add the search strategy used, a paragraph on the strengths and limitations of the review and the definition of abbreviations in Table 1 and in the figures

Author Response

Reply: Thank you very much for these constructive comments. Accordingly, we’ve added the search strategy used, the content on the strengths and limitations of the review and the definition of abbreviations in Table1 and in Figure 2 as following:

  1. Page 2 line 14-19, in revised manuscript:

We’ve added the search strategy used— “Therefore, we conducted a literature review to examine the role of saponins in the treatment of neuropathic pain. The search was performed in the PubMed and CNKI databases with the following keywords: saponins and neuropathic pain or pain. Relevant articles, including research papers, reviews and their selective references, were examined systematically and are cited in the present review.”

  1. Page 13 line 46 to Page 14 line 2, in revised manuscript:

We’ve added the content on the strengths and limitations of the review—

“5. Strengths and limitations of the review

The strengths of this review are to systematically and comprehensively summarize the effects and mechanisms of different saponins in neuropathic pain. At the same time, according to the current research technology and progress, some limitations of the previous research are pointed out and some constructive opinions are put forward.

On the other hand, this paper also has a disadvantage. This review is based on the structure and type of saponins, which leads to the inability to systematically summarize the role of saponins in different types of neuropathic pain, which only be introduced separately in each part.”

  1. Page 6 below Table1 and Page 7 in Figure 2, in revised manuscript:

  We’ve added the note as the definition of abbreviations, below Table1— “Note:TSE, total saponin extract; TNI, tail nerve injury; SCI, spinal cord injury; CCI, chronic constrictive injury; DRG, the dorsal root ganglion; AITC, allyl isothiocyanate; STZ, streptozotocin; LPS, lipopolysaccharide; BFEO, Bupleurum falcatum L. roots essential oil.”

in Figure 2— “(Note: PPD, protopanaxadiol; PPT, protopanaxatriol; Glc, glu-coside; Rha, rhamnoside; Fuc, fructoside)”.